# Social Media User Behavior and Emotions during Crisis Events

**DOI:** 10.3390/ijerph19095197

**Published:** 2022-04-25

**Authors:** Mingyun Gu, Haixiang Guo, Jun Zhuang, Yufei Du, Lijin Qian

**Affiliations:** 1College of Economics and Management, China University of Geosciences, Wuhan 430074, China; gmingyun@cug.edu.cn (M.G.); 20181001343@cug.edu.cn (Y.D.); qianlijin@cug.edu.cn (L.Q.); 2Research Center for Digital Business Management, China University of Geosciences, Wuhan 430074, China; 3Mineral Resource Strategy and Policy Research Center, China University of Geosciences, Wuhan 430074, China; 4Department of Industrial and Systems Engineering, School of Engineering and Applied Sciences, University at Buffalo, SUNY 317 Bell Hall, Buffalo, NY 14260, USA

**Keywords:** social media, information dissemination, sentiment analysis, COVID-19

## Abstract

The wide availability of smart mobile devices and Web 2.0 services has allowed people to easily access news, spread information, and express their opinions and emotions using various social media platforms. However, because of the ease of joining these sites, people also use them to spread rumors and vent their emotions, with the social platforms often playing a facilitation role. This paper collected more than 190,000 messages published on the Chinese Sina-Weibo platform to examine social media user behaviors and emotions during an emergency, with a particular research focus on the “Dr. Li Wenliang” reports associated with the COVID-19 epidemic in China. The verified accounts were found to have the strongest interactions with users, and the sentiment analysis revealed that the news from government agencies had a positive user effect and the national media and trusted experts were more favored by users in an emergency. This research provides a new perspective on trust and the use of social media platforms in crises, and therefore offers some guidance to government agencies.

## 1. Introduction and Literature Review

Social media platforms have allowed users to become the main force behind information creation and dissemination and are also now widely used for knowledge discovery related to daily news [1,2] and emergencies and crises [3]. Recent studies have found that social media data is being harnessed for disaster emergency management [4,5,6], crisis perception analyses [7,8,9], sentiment analyses [10,11,12], and rumor analyses [13,14,15]. Therefore, the massive social media data generated during emergencies is being employed to assist in emergency management. Integrating social tools into disaster preparedness activities can promote professional emergency personnel and citizens to use familiar tools to make effective emergency response during crisis [16]. Social media tools have low cost, wide coverage, and proven advantages before, during, and after the crisis [17].

Due to the convenience and openness of social media, governments are now harnessing the power of social media to communicate with their citizens [18,19]. In particular, when events highlighted through social media threaten government credibility, timely and effective announcements by the authorities can minimize public doubt and calm public emotions. Social media is now being widely used to communicate emergency information to the general public [20] and has therefore become a vital part of public risk management during emergencies, with most government agencies using social media to clarify rumors, issue warnings, and improve service efficiency [20,21,22]. However, generally, government agencies have tended to ignore the interactive nature of social media, only using the platforms to publish information [23] or interact with a small number of stakeholders [24]. While early studies have highlighted the importance of social media government interactions with the public, most governments have been slow to change. Many social media platforms such as Twitter and Sina-Weibo have introduced a “verification” mechanism; for example, Twitter’s blue verification badge and Sino-Weibo’s institutional or personal certifications alerts readers to the authenticity of the information [25]. Official government departments and national media accounts have also become important channels for the release of government information during a crisis or to calm public emotions.

There has been a significant increase in studies focused on information dissemination during emergencies to determine the information transmission influencing factors [26,27], understand the social media information coverage rates and communication efficiencies [14], and analyze user statuses during the crisis events [28]. As fake news often has more novel content than real news, it tends to spread more widely [29]; therefore, studies have used time series methods to explore information dissemination, finding that online public opinion communication had a comprehensive four stage process: a potential stage, an emergency stage, a communication stage, and a resolution stage [30]. Before a crisis enters the emergency period, it has been found that the social platform content changes from being an information medium to a channel for emotion and fear transmission. Ref [31] examined information diffusion based on network structure and time, and proposed a discrete-time bi-probability independent cascade model. Based on the reposting behaviors on Sina-Weibo during the 2012 Yiliang earthquake event, Li et al. (2018a) developed a content-based, multi-category Naïve-Bayesian classifier that divided the microblog information posts into five categories and then analyzed the transmission patterns in these five categories during the different post-earthquake stages. A recent study on the information spread during the COVID-19 epidemic found that there were more tweets containing false information but fewer reposts than tweets based on scientific evidence or fact verification, which were in turn more attractive than simple facts [32]. Social media has also been found to be an effective medium for communication between the public and the authorities during a crisis [33], with official organizations using social media platforms to communicate warnings and risks and resolve rumors [20]. However, there has been little research on user responses to official news and the impact of this news on user emotions.

Previous studies have tended to focus on the stimulating effects of emergencies on the activities of social media network users; however, there has been little research on the effects the social media-posted messages have on the behaviors and emotional responses of users during emergencies. Therefore, to go some way to filling this research gap, this study examined social media user behavioral and emotional responses to messages from verified accounts.

Wuhan, Hubei Province had been monitoring influenza and related diseases in December 2019 and had identified several cases of viral pneumonia, all of which had been diagnosed as viral pneumonia/lung infection. On 30 December 2019, Dr. Li Wenliang announced in a clinical 04 class group at Wuhan University that “seven cases of SARS have been confirmed in the seafood market of South China” and reminded fellow clinicians to “pay attention to their families and relatives”. An hour later, he added that “the latest news is that a coronavirus infection has been confirmed and virus typing was under way.” On 3 January 2020, he was warned and admonished by the local police station for “making false statements on the internet”. Dr. Li Wenliang was consequently diagnosed with COVID-19 on 31 January (posted on 1 February through his Weibo), and an official report on 7 February confirmed that Li Wenliang had died at only 34 years old. On 7 February 2020, with the approval of the Central Committee, the National Supervisory Commission decided to send an investigation team to Wuhan, Hubei Province, to conduct a comprehensive investigation into the problems reported by the public involving Dr. Li Wenliang. This event quickly attracted the wide attention from users on the Chinese social media platforms Sina-Weibo, WeChat, Tik-Tok, and Zhihu, with many people expressing their deep condolences for Dr. Li’s misfortune and raising questions about the government’s actions. Table 1 summarizes the top 10 focal topics in chronological order, with the entire event timeline shown in Figure 1.

In particular, events related to “Dr. Li Wenliang” during the COVID-19 epidemic in China were selected as the research object. The most important reason why we chose the incident related to “Dr. Li Wenliang” was its long duration and the intense discussion on social media in China during the early days of COVID-19 in Wuhan. As this event has also attracted attention in the international community, the comparison of user behavior and emotion of social media platforms in different countries is added in the research process to find the differences of users’ attitudes towards the same event in different countries. So, we decided to choose this topic to answer the following research questions:

(RQ1) What are the differences between the institution-verified and personal-verified account foci during a trust crisis?

(RQ2) What are the differences in the emotional expressions used in these different social accounts?

(RQ3) What are the differences between the messaging capabilities of the different social accounts?

The remainder of this paper is organized as follows. Section 2 introduces the data sources and methodology, Section 3 analyzes the results, Section 4 discusses the analysis results, and Section 5 concludes the paper and provides possible future research directions.

## 2. Data Collection and Methodology

The social media discussions on Sina-Weibo about “Dr. Li Wenliang” during the COVID-19 outbreak in Wuhan were selected as the research object because this discussion thread attracted widespread enthusiastic attention on social media platforms over a fairly long duration and therefore provided sufficient research data, the government agencies issued multiple fact sheets in association with this incident which provided valuable research samples, and this matter had a complicated development and underwent many reversals, which allowed for a detailed analysis of the changes in the user behaviors and emotions. According to Sina-Weibo, the subjects associated with the event generated 239,000 discussions and 760 million views. In this process, the government and the public have been in a state of communication on social platforms. The government disclosed relevant information, the public expressed their views on the government’s statement, and the government disclosed more information according to the public’s response. The topic discussion about “Dr. Li Wenliang” has been hot on Sina-Weibo for nearly one month.

Sina-Weibo, which is one of the most popular social media sites in China, is similar to Twitter as users can receive news and information, participate in discussions, and express their emotions. Sina-Weibo’s monthly active users were reported to have reached 516 million in 2019, with mobile accounts accounting for 94%. With 222 million daily active users, 22 million more than in 2018, the endless stream of data had made Sina-Weibo an important data source for social media research. The Sina-Weibo platform has many different topics which users can respond to using microblogs. Leveraging the Sina-Weibo hashtag function and Sina-Weibo API, all Sina-Weibo posts under the focal topics summarized in Table 1 from 6 p.m. on 1 January 2020 to 12 p.m. on 1 March 2020 were captured. The data obtained in this paper can be found publicly on the Sina-Weibo platform, and all the data that can identify the user’s identity are filtered out in the process of capturing the data. There are no personal data displayed in the paper. The analysis is also analyzed from the perspective of the whole community, without infringing on personal privacy

We first cleaned and processed all the data collected through microblog API. We collected data from multiple dimensions, including keywords, user names, microblog content, publishing time, release tools, etc. Considering the research of this paper, we finally took the keywords, user name, microblog content, number of forwarding, number of comments, number of likes, publishing time, authentication, whether it is forwarding microblog, and the original microblog content of each microblog as the main research objects. The first step was to eliminate invalid data. A total of five students participated in data cleaning according to the unified deletion standard. According to the research methods in the previous literature, we adopted the following principles to complete the work of data cleaning [34,35]. The deletion criteria were as follows: (1) meaningless content, such as “the original microblog has been deleted”; (2) content unrelated to Dr. Li Wenliang, such as some bloggers’ daily interaction; (3) random code; (4) duplicate content. Through manual identification, microblogs that have been deleted, cannot be viewed, and have nothing to do with topics were regarded as invalid data. If the original microblog is related to the topic and the content is not expressed when forwarding, it was also retained. After data cleaning, the filtered invalid data were deleted, which, after removing the non-related messages totaling 190,627 posts, is shown in Table 2, with the corresponding daily trends being shown in Figure 2. In order to compare with Sina-Weibo data, we also collected 44,309 tweets from the Twitter platform for sentiment analysis.

In order to carry out the follow-up emotional analysis, we preprocessed the effective data. Firstly, we mainly divided the data into forwarding microblog and original microblog, and in the forwarding microblog data, we divided it into forwarding content microblog and no-content forwarding microblog. For the definition of no-content forwarding, it refers to the content that users do not express their opinions when forwarding microblogs, because the data volume of no-content forwarding is relatively large, and it has the meaning of analysis, which means to support the content of the original microblog and classify its emotion as the same as that expressed by the original microblog, we decided to overlay the content of no-content forwarding microblog with the original microblog content, and preprocessed the microblog content, delete the @ ID name, microblog topics (##), and other information that may interfere with microblog emotion.

For the processed microblog data, the next step was emotional classification. Before the training algorithm, the emotional labels of some training data should be labeled manually—in order to ensure the consistency of classification results, this part of the work was completed by one person independently. We selected 34% of the total daily microblogs for annotation, of which 45% were microblogs with forwarded content and 55% were microblogs without forwarded content (including original microblogs) in order to ensure the content covered by the data of the annotation set was more comprehensive. The proportion here was consistent with the proportion in the total microblogs. We divided the emotions expressed in microblog content into three kinds: positive emotions, neutral emotions, and negative emotions. The judgment of positive emotions was that the emotions and attitudes revealed in the microblog content are positive and reflect the optimistic side, such as moving, optimistic, praise, and other emotions. The microblog content marked as neutral emotion was mainly the microblog content that does not clearly disclose their emotions and attitudes, such as some press releases related to the event content, discussion of objective facts, etc. The last is the judgment of negative emotions. For some microblog content with satire, pessimism, anger, abuse and other bad emotions, we classified it as negative emotions.

ROST Content Mining 6 software (hereinafter referred to as ROST CM6 software) was employed for the word segmentation. First, the preprocessed data file was converted into a .txt file, after which word segmentation processing was performed on the text content using the word segmentation functional analysis in ROST CM6 [36]. To improve the accuracy, the skip-gram model [37] was then used for the vector construction to generate additional training samples and elicit more semantic detail from between the words.

Support vector machines (SVMs) are a type of supervised machine learning method which is often used for classification. It has a solid theoretical foundation and has been developed rapidly since 1990s and has derived a series of improved and extended algorithms, which is also a common machine learning method for classification that has been widely used for text classification [38] and sentiment analysis [11,39]. Many researchers have proposed that support vector machine may be the most accurate text classification method [40]. In this paper, we first carried out text processing, feature selection, and data normalization, and then achieved the emotion classification model through SVM training. This paper uses the open source LIBSVM model to avoid the instability of development. In the training process, we used the cross-validation parameter optimization method, and selected RBF as the kernel function through experiments, the data were then applied for the sentiment tendency discrimination to divide the sentiment messages into positive, negative, and neutral labels (Figure 3). Python and MATLAB programming languages were used for the data processing.

The ROC (Receiver Operating Characteristic) curve is recognized as the most rational choice for imbalanced data, which depicts relative trade-offs between the benefits and costs (Fawcett, 2006). The threshold of a classifier presents the degree to which an instance is a member of a class. In practice, we often use the Area Under ROC Curve (AUC) as a scalar measure instead of ROC curve—the larger the AUC value, the better. The accuracy of our experiment measured by AUC is shown in Figure 4. In the process of emotion classification, we adopted binary classification methods to classify the experimental data. Firstly, the microblogs with positive emotions were separated from other microblogs, then the microblogs with negative emotions in the remaining microblogs were separated, and finally the remaining microblogs were taken as neutral attitude. Since there were a large number of microblogs with neutral emotions after the first classification, we classified them again. The data results after the two classifications were combined for subsequent analysis. In this study, we leveraged the “Sklearn” package, a very powerful machine learning library provided by a Python third party, to measure the accuracy of the algorithm. We imported “Roc_curve” and “Auc” functions from the “Sklearn.metrics” module, used “Roc_curve” to calculate the TPR (True Positive Rate) and FPR (False Positive Rate) of the test set, then used “Auc” to calculate the value of AUC, and finally drew the ROC curve. The four ROC curves in Figure 4 show the accuracy of four classification processes in two classifications.

## 3. Data Analysis

### 3.1. Comparison of the Verified and Non-Verified Accounts

The Sina-Weibo platform has three main user types; institutional-verified accounts, personal-verified accounts, and non-verified accounts. The institutional-verified accounts include the official media, enterprises, and public/government institutions, the personal-verified accounts include actors, athletes, and entrepreneurs, and the larger number of non-verified accounts are personal users. As shown in Figure 5, of the 190,627 messages examined during the research period, 63% (119,530 of 190,627) were from verified accounts (41% institutional-verified, 22% personal-verified) and 37% were from unverified accounts. The institutional-verified account messages received 1,233,013 forwards, 1,317,919 comments, and 13,049,620 likes, the personal-verified accounts received 365,010 forwards, 251,714 comments, and 2,591,215 likes, and the unverified account messages received 194,480 forwards, 100,602, comments and 902,813 likes.

With 6 h as the time unit, Figure 6 shows the changes in the number of messages published by the different account types from 1 January 2020 to 1 March 2020, from which it can be seen that the publishing activities of all three types were similar over time, with the discussion reaching a peak on 1 February and 7 February, during which time there were many focal events. The analysis of the message sources revealed that most of the 10 focal topics had been initially published by institutional-verified accounts, such as @People’s Daily, @CCTVNews, @WuhanPolice, and @PeerVideo. For example, @CCTVNews posted the message “With the approval of the Central Committee, the National Supervisory Committee decided to send an investigation team to Wuhan, Hubei Province to conduct a comprehensive investigation on the relevant issues of Dr. Li Wenliang reported by the public”, which received 186,000 forwards, 150,000 comments, 1,950,000 likes, and aroused extensive discussion. Although institutional-verified accounts were often the source of the focal topics, Figure 6 reveals that the unverified accounts tended to generate longer discussion durations, with the verified accounts being more active in the early topic generation stage, and the personal-verified accounts being the slowest to respond to new trending topics.

The analysis of the interactions between the three Sina-Weibo accounts found that the messages from the verified accounts, whether they were forwarded, commented, or liked, played a more significant public communication role than the unverified accounts. Figure 7 shows the cumulative interaction distributions between the verified and unverified accounts.

### 3.2. Sentiment Analysis

#### 3.2.1. Sentiment Analysis of Sina-Weibo Data

An emotional classification and a timeline trend analysis of the developments related to the “Dr. Li Wenliang” event in the 190,627 messages were conducted based on the focal topic public opinion developments by the different user types. The user messages were divided into positive, negative and neutral categories and as few people had paid much attention to “Dr. Li Wenliang” before 29 January 2020, the change trends in the three emotions were tracked from 29 January 2020 to 1 March 2020. Figure 8 shows the proportion of three emotions per day from 29 January 2020 to 1 March 2020, from which it can be seen that the negative emotions were sustained for a longer time in the development of public opinion. When the “Dr. Li Wenliang” topic first arose, most users expressed dissatisfaction; however, when the official agency verified account @PeaceWuhan issued a timely announcement in response to the incident on 30 January, there was a rapid decline. Other institutional-verified accounts immediately issued many messages and the expert Zeng Guang evaluated Dr. Li Wenliang positively in an interview with CCTV (China Central Television) @CCTVNews, which appeared to be an official attempt to calm things down, with an interview from the national press and a government response also being published.

The second wave of negative sentiment came on 6 February 2020, when rumors spread on Sina-Weibo that Dr. Li Wenliang had died; however, the authorities did not respond. Consequently, there was a perception that the government was hiding the truth, which increased the negative feelings. Subsequently, in the early hours of 7 February 2020, @WuhanCentralHospital, an institutional-verified account, announced Dr. Li Wenliang’s death and @CCTVNews announced that the State Supervisory Commission had sent an investigation team to investigate Li’s problems. Similar to the previous occurrence, there was a flurry of messages published from institutional-verified accounts, which slowly turned the public sentiment to sympathy and condolences for Dr. Li Wenliang. However, in the following days, the public’s negative sentiment continued to increase, and on 12 February 2020, many institutional-verified accounts reposted the video in which academician Zhong Nanshan had praised Dr. Li Wenliang as a hero. As Zhong Nanshan was trusted by the public, his views led to a positive change in the public sentiment, after which there were no more discussions on the focal Dr. Li Wenliang event. With the decrease in the information released by the verified accounts, the overall mood was negative. It appeared that the verified accounts, and especially the government agencies’ timely responses to public queries, had a positive effect on reversing the public sentiment. The posting of many messages at the same time by the verified accounts calmed public sentiment, which had been the government’s intention. Although this practice affected public opinion in the short term, over the long run, the effect lessened.

Overall, there were significant differences in the emotions expressed by the three accounts throughout the event. The verified accounts posted mostly positive messages, whereas the unverified users expressed strong negative emotions (Figure 9). Figure 10 shows the various emotional proportions for the three account types for the different focal topics, from which it can be seen that the verified account emotions varied little and tended to be mainly positive and the unverified account emotions also varied little but tended to be negative across almost all topics, with most unverified users being very clear about their attitudes and clearly regarding Sina-Weibo as a platform for emotional expression. However, while the verified accounts expressed both positive and neutral views, these were mainly associated with facts as Sina-Weibo’s main function was seen as information/news dissemination.

#### 3.2.2. Sentiment Analysis of Comments

In order to further verify people’s attitude towards information released by official accounts, we selected 29 January to 10 February 2020 as the research period, which has the most popular public opinion. We selected 31 popular microblogs from 17 official certified accounts, and analyzed the popular comments under each microblog, we selected 7454 popular comments from the 31 popular microblogs, which received 1,048,936 likes and 77,580 comments (Table 3). We have classified all the comments by emotion. Unlike the original microblog, comments that clearly express their own attitude are more likely to be recognized by people (Figure 11).

We chose the microblogs that receive the most positive comments and the most negative comments to compare. The topics that received the most positive comments were “Zhong Nanshan wept when talking about Dr. Li Wenliang”, “Citizens presented flowers to mourn Dr. Li Wenliang”, and “Dr. Li Wenliang diagnosed with COVID-19”. The main themes in the comments were respect, appreciation, and prayers for Dr. Li Wenliang. The microblogs that received the most negative comments were “the report on the parents of Dr. Li Wenliang” and “the official news about the death of Dr. Li Wenliang”. People expressed their dissatisfaction with the government under these microblogs, hoping to investigate the responsibility of government staff and sympathize with Dr. Li Wenliang (Figure 12).

Through comparison, we can clearly find that although the news reported is similar, because people attribute the death of Dr. Li Wenliang to the dereliction of duty of the government and target the government’s previous wrong behavior, therefore, the comments on official media tend to show negative emotions, while reports on ordinary media tend to express sadness. It was not until @People’s Daily, which received the most likes, made it clear that the National Supervisory Commission would investigate Dr. Li Wenliang that these emotions became positive, which also reflects the importance of official media responding to ordinary netizens’ demands in a timely manner.

#### 3.2.3. Sentiment Analysis of Twitter Data

In addition to the emotional analysis of the data from Sina-Weibo, we also captured the data from Twitter, a more widely used platform in the world, for comparative study. We collected 44,309 tweets with the keyword “Li Wenliang”. In order to better understand the real views of the people in other countries on the “Li Wenliang” incident, we divided all tweets into “Chinese” and “Non-Chinese” categories. After statistics, the number of two analogies was 25,607 and 18,702, respectively (Figure 13). Figure 14 shows the overall sentiment classification results and the sentiment analysis of the two analogues. In all the tweets published in Chinese, negative emotions dominate, which is similar to the emotional expression of the unverified accounts of Sina-Weibo. Due to the Internet restrictions on twitter in China, it is safe to assume that think that the overseas Chinese are basically in line with the domestic people on this matter. In all Non-Chinese tweets, neutral emotion is dominant, and its emotional expression is similar to that of verified users on Sina-Weibo. By looking at these tweets, we found that most of them were media reports, so the emotion is relatively objective. In the positive mood, many people expressed their prayers and sympathy for Dr. Li Wenliang, while in the negative part, they expressed more distrust of the government and anger at the death of Dr. Li Wenliang.

When analyzing the data of Sina-Weibo platform, we also used Area Under ROC Curve (AUC) to evaluate the classification accuracy in this part, as shown in Figure 15, which is the accuracy of our classification test on Twitter data.

### 3.3. Impact of the Verified and Non-Verified Accounts

According to China’s constitution, it is a crime for rumors in social media to be forwarded more than 500 times. Therefore, when analyzing Sina-Weibo data, we generally believe that the messages forwarded more than 500 times are true and reliable, which is widely recognized by the public. Of all the messages, only the 0.15% that were forwarded more than 500 times were analyzed (Table 4). Overall, message originality appeared to be crucial as to whether users were willing to repost them. The analysis revealed that the verified accounts were more likely to be forwarded, which confirmed the value of these accounts in spreading news during a crisis. The ratio of the number of messages to the number of accounts was counted from which it was found that the messages published by the same verified accounts were more likely to be forwarded multiple times, a function not available to unverified users. Message videos, pictures, or emojis were not found to have any obvious connection with multiple reposts; however, the inclusion of a picture in the unverified user messages appeared to have certain advantages. Combined with the sentiment analysis results, it seemed that the pictures were helpful in expressing the message sentiments.

To exemplify the differences in the message spread from the different accounts, four messages that had similar forwarding times were selected for comparison from six different accounts; @People’s Daily, @ZiGuangGe @CCTVNews, @GuDaBaiHua, @AKindOfIdealRepooter, and @FlyFish12341234; each of which was forwarded 1652, 1210, 1981, 1480, 1236, and 746 times, respectively. In Figure 16, the nodes represent the Sino-Weibo users, and the lines between nodes represent the connectivity through the attribute mentions. The size of the nodes (user name) indicates the number of times a user was mentioned, and the different node and edge colors indicate the different clusters. As can be seen, the government agency account announcements from @People’s Daily and @ZiGuangGe (Figure 16a,b) spread in one direction and did not form a new dissemination center after the spreading, indicating that the value of these kinds of messages was in the transmission of information. The news media account @CCTVNews (Figure 16c) spreads news and triggers discussions; therefore, there were more people participating in the discussion and multi-directional communication. The @GuDaBaiHua, @AKindOfIdealRepooter and @FlyFish12341234 (Figure 16d–f) are personal accounts; however, as @GuDaBaiHua and @AKindOfIdealRepooter (Figure 16d,e) are personal-verified accounts, their message spreads resulted in several new discussion centers, possibly because personal-verified accounts usually have no discussion restrictions, that is, after a focal message is generated, it is forwarded by other similar personal authentication accounts. The difference between these is in the number of new discussion centers, which could be related to the differences in the number of fans on their accounts (@GuDaBaiHua with 13 million followers and @AKindOfIdealRepooter with 5 million followers). The unverified account transmission was found to be similar to the institutional-verified accounts as they tended to simply trigger reposting without the development of additional discussions, with other users only expressing their agreement with the point of view (Figure 16a,b,e,f). As the users participating in the reposting were ordinary users, their followers and communication capabilities were generally limited, making it more difficult to generate wider discussions and dissemination.

The themes in the messages that had more than 500 forwards were analyzed by account type and the three top-ranked themes selected, two researchers participated in the work of thematic analysis. In this chapter, Cohen’s kappa coefficient is used to measure the consistency of the evaluation of the two researchers, and it is found that the two raters have achieved substantial consistency (k = 0.67). With the research being similar to the results from the previous analyses, that is the verified accounts focused more on disseminating news and announcements and the unverified users were more concerned with event development and were more likely to use emotional messages (Table 5).

## 4. Discussion

This paper focuses on the research on the differentiation of social media users’ behavior and emotion under the background of public emergencies. Section 4 studies and analyzes the behavior differentiation, emotional differentiation, and influence differentiation of social media users. According to the results of Section 4, this section puts forward the following suggestions for the control of social media public opinion at the individual level in the context of public emergencies:

(1) Pay real-time attention to public opinion hotspots guided by ordinary users.

The research in Section 4 shows that the discussion hot spots led by ordinary users continue to spread on the social media platform for a longer time. Combined with the results of emotional analysis, negative topics are often the focus of social media users’ attention. Timely and effective supervision of public opinion hot spots: on the one hand, refute rumors immediately when rumors arise to curb their spread; on the other hand, guiding the focus of social media users timely and actively responding to the focus of public opinion plays a positive role in reducing the overall negative emotions of social media platforms.

(2) Enhance the “interactivity” between official media users and ordinary users.

Through the comprehensive analysis of the “Dr. Li Wenliang” incident, it is not difficult to find that the certified accounts, especially the timely response of government agencies to public inquiries, have had a significant positive impact on reversing public sentiment. In the context of public emergencies, official media users should not only pay attention to the ability of social media platforms to spread information, but also strengthen their own “sociality”, open dialogue channels with ordinary users and expand dialogue methods with ordinary users, which is of great significance to improve the mood of users of social media platforms.

(3) Pay attention to the role of opinion leaders in emotional communication.

Throughout the development of the “Dr. Li Wenliang” incident, two important opinion leaders spoke, one was “ Zeng Guang responded that eight people were interviewed in Wuhan “, the other was “Zhong Nanshan shed tears when talking about Dr. Li Wenliang”. Two top scholars in the professional field played an important role as opinion leaders on the social media platform. The research results in Section 4 also fully prove the good effect of these two speeches on the emotional guidance of users of social media platform. Their authority and influence ensured their speeches achieved a high communication effect, and guided the positive feelings of ordinary users while achieving information diffusion. The occurrence of public emergencies is inevitable, but the guidance of authoritative and professional opinion leaders to public opinion plays a vital role in curbing public opinion crisis.

(4) Timely and public response to the voice of the masses.

In the early stage of public emergencies, all kinds of news on social media platforms emerge endlessly, with different opinions, and the lack of ability to judge the authenticity of information is easy to breed negative emotions on social media platforms. The research on popular comments in Section 4 shows that social media platform users’ express dissatisfaction, accusations, and other negative emotions under the news released by officially certified platform users, which is, to some extent, the people’s appeal to the government. Regulators should pay attention to such negative public opinion and respond to public dissatisfaction in a timely manner. Never be invisible, or even delete comments and close the comment area, so that negative emotions accumulate and even derive into vicious public opinion events, leading to a greater degree of trust crisis.

(5) Timely start to accident accountability.

The public opinion event of “Dr. Li Wenliang” is due to the dereliction of duty of some government staff, which has also become the fundamental reason for users of social media platforms to vent their emotions. In case of such incidents, the official should start the accident accountability procedure “seriously, quickly and strictly”, and give full responsibility to the ability and influence of officially certified users to disseminate information on social media platforms. The results of the user emotion analysis in Section 4 show that “the investigation team appointed by the state supervision to investigate Li Wenliang’s problem” is an important time node for the turning point of the “Dr. Li Wenliang” event. At the same time, the research in Section 4 also confirms the ability of officially certified users to spread information on social media platforms. Based on this, the research believes that “timely processing and rapid release” is an important time for the government in the face of trust crisis, an important means to quell negative public opinion on social media platforms.

## 5. Conclusions and Future Research Directions

This paper examined a popular social media topic in China during the COVID-19 crisis to analyze the behavioral and emotional changes in different Sina-Weibo platform account types, the specific interactions, and the possible impact on the behavior and emotions of the unverified account users. It was found that the focal topic origins often came from institutional-verified accounts, with official news, briefings, or expert interviews tending to attract greater attention. However, similar to the results in previous studies, official agencies were found to primarily use the social media platform as a medium to publish information rather than to interact with ordinary users [22], which meant that their participation in the topic discussions was short. However, the unverified account users were found to be more willing to spend time participating in the discussion and more often used Sina-Weibo as a platform to express their opinions and emotions. Compared with the other two account types, the personal-verified contribution of the focal events was not so positive, possibly because their user identities restricted their ability to comment on sensitive events. The verified accounts were found to have better news dissemination capabilities. In uncertain circumstances, the most focused on messages dealt with official news, the analysis of which confirmed previous research that media richness was negatively correlated with user participation [41].

The differences in emotional expression between the different user types were also explored, from which it was found that positive comments dominated the messages posted by the verified accounts, while the unverified accounts were more likely to express negative emotions. The “Dr. Li Wenliang” incident caused the public to question the credibility of the government, with nearly all focal topic discussions being negative. However, for the topic “#*The State Supervision Commission sent an investigation team to investigate Dr. Li Wenliang#*”, most unverified users had positive attitudes, which indicated that timely and effective government responses to public doubts can appease the public mood. The analysis of the emotional changes found that the emergence of new hot topics resulted in changes in user emotion, and that government agency news had a positive effect on user emotions, which suggests that the public continued to have a positive attitude towards the government in the face of uncertainty. It was therefore concluded that when public credibility is questioned, it is necessary for government agencies to release information in a timely and effective manner.

During the crisis, the most retweeted Sina-Weibo posts were mostly from verified accounts, with news from the national media being the most interesting, followed by government agencies, and then the ordinary media, which was also confirmed in the analysis of the three account type messaging capabilities. The content analysis revealed that the national media and government agencies were not overly emotional, but that the general media tended to convey certain emotions in their posts. In the personal-verified COVID-19 content, information from technology and professional knowledge accounts was found to attract more attention, with verified account posts being more likely to be reposted.

The findings from this analysis could be helpful for government agencies dealing with similar crises of confidence. First, it is essential that government agencies communicate effectively and promptly when a crisis occurs as this study revealed that Sina-Weibo users tended to obtain their information from verified accounts. Therefore, government agencies could cooperate with official media, experts, and scholars who have high public trust to release information in various forms to respond to the public’s questions and calm their emotions. Similarly, government agencies could maintain a certain continuity on their social media platforms to assess and maintain public emotions. This study provides new ideas for the development of verified accounts and provides government agencies with new ideas for the use of institutional-verified accounts in trust crisis events.

However, this study had certain limitations. First, only the number of forwards, comments or likes were counted and any new data arising from these social media activities was not examined. Further questions could also be explored, such as, in the event of a crisis of confidence, how can the authorities better use social platforms to spread information and appease public sentiment, and what government agency actions are needed to reverse the public’s negative emotions? While this paper only compared the information dissemination capabilities of different accounts, the research could be extended to examine how the messages from the verified accounts changed the behaviors and emotions of the unverified accounts, which variables were most affected, how these changes affected the behavior of the verified accounts, and how the verified accounts could eliminate any negative impact. It would also be interesting to analyze more multi-source data to assess the possible impact of messages linked to external news sources on the social media news dissemination.

## Figures and Tables

**Figure 1 ijerph-19-05197-f001:**
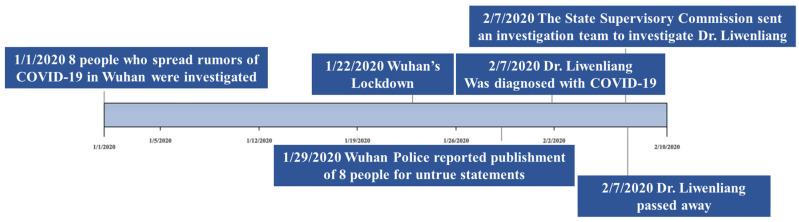
Timeline of events related to “Dr. Li Wenliang”.

**Figure 2 ijerph-19-05197-f002:**
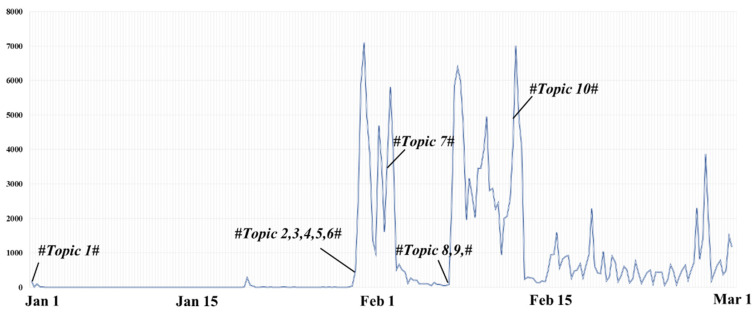
Trends for the messages of Sina-Weibo related to “Dr. Li Wenliang”.

**Figure 3 ijerph-19-05197-f003:**
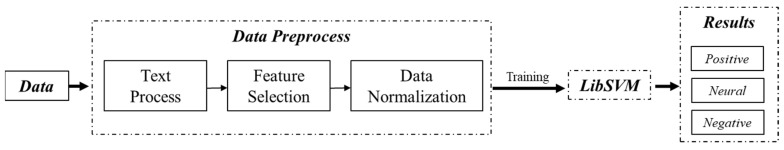
Emotion classification process.

**Figure 4 ijerph-19-05197-f004:**
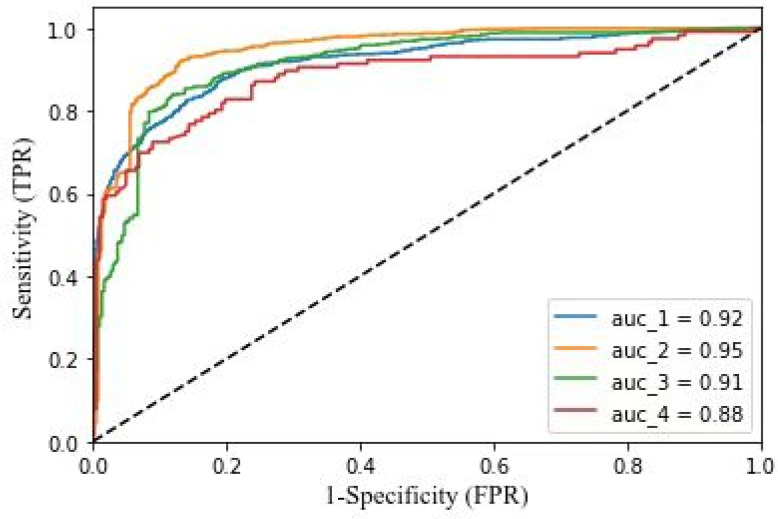
AUC diagrams of the algorithm.

**Figure 5 ijerph-19-05197-f005:**
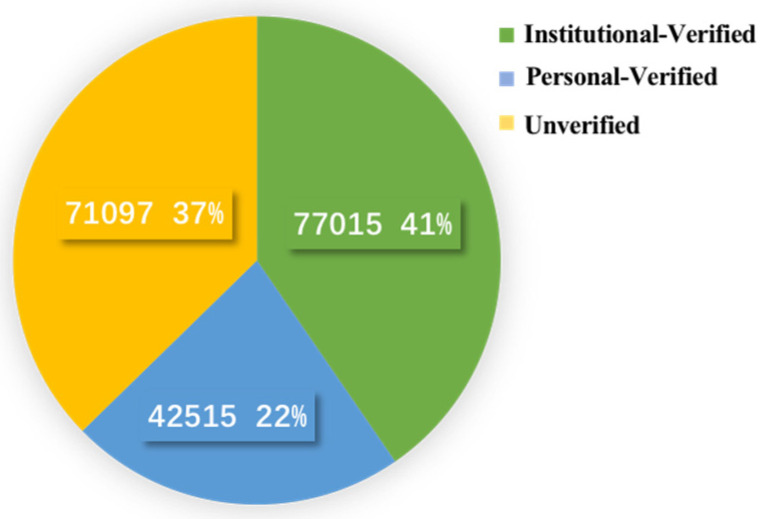
Distribution of the number of messages posted by the three account types.

**Figure 6 ijerph-19-05197-f006:**
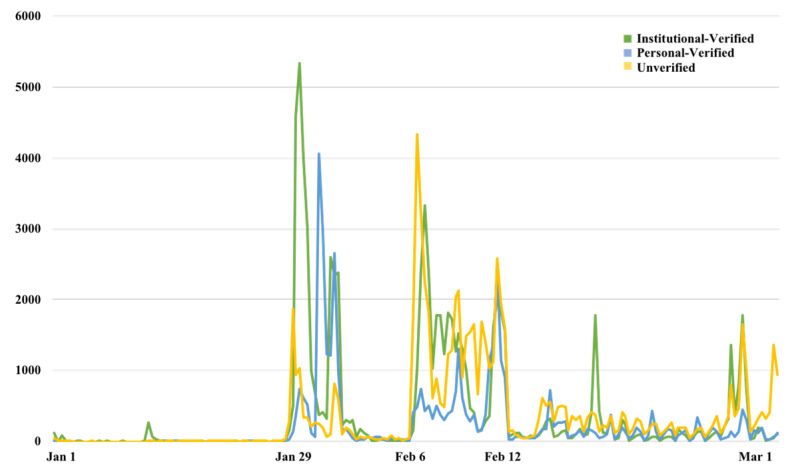
Number of messages posted by the different account types.

**Figure 7 ijerph-19-05197-f007:**
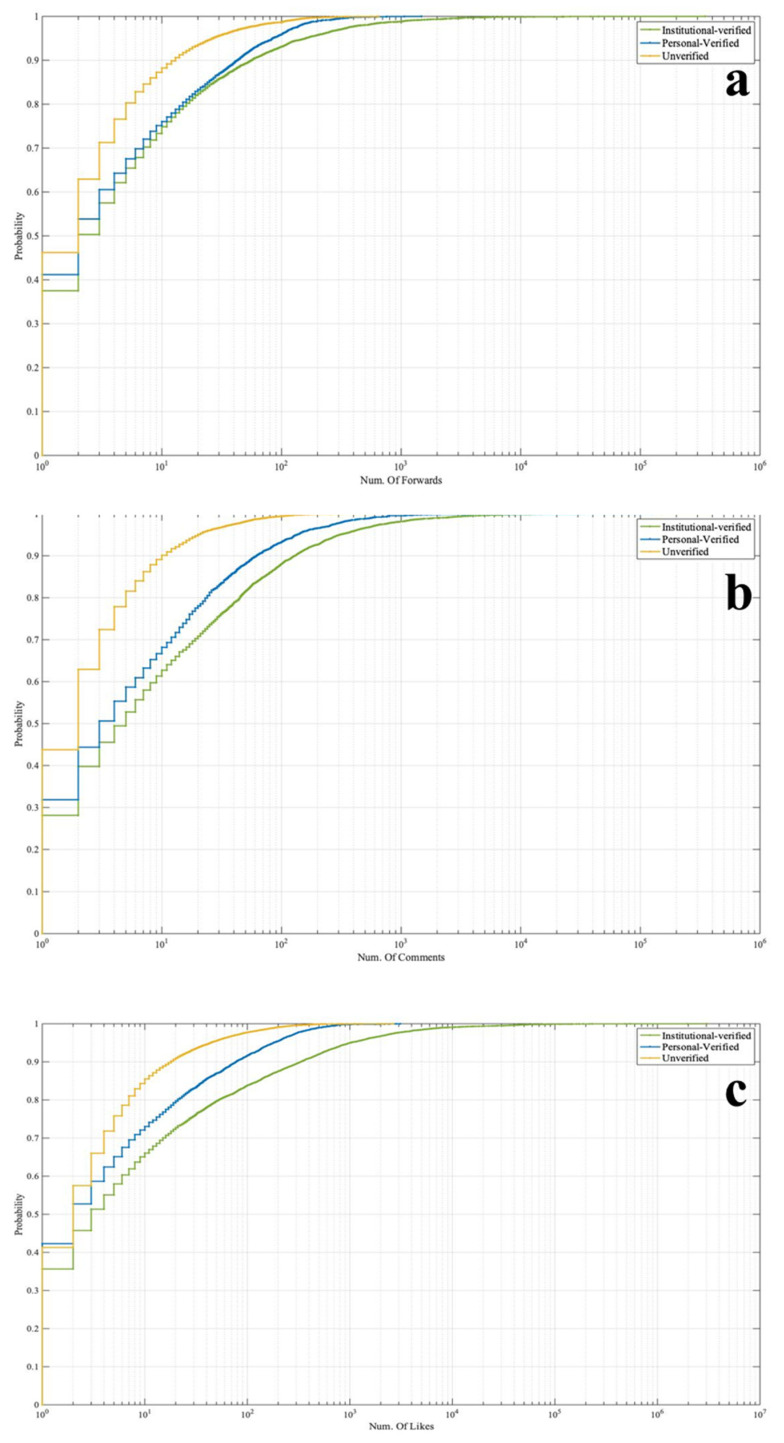
CDF for the Forwards (**a**), Comments (**b**), Likes (**c**) from the different account types.

**Figure 8 ijerph-19-05197-f008:**
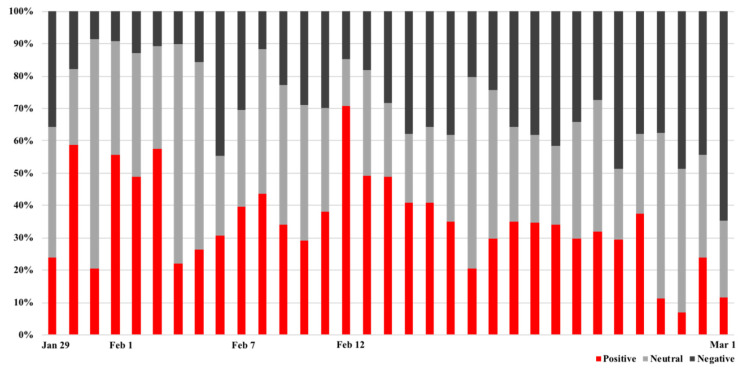
Emotional changes from 29 January to 1 March.

**Figure 9 ijerph-19-05197-f009:**
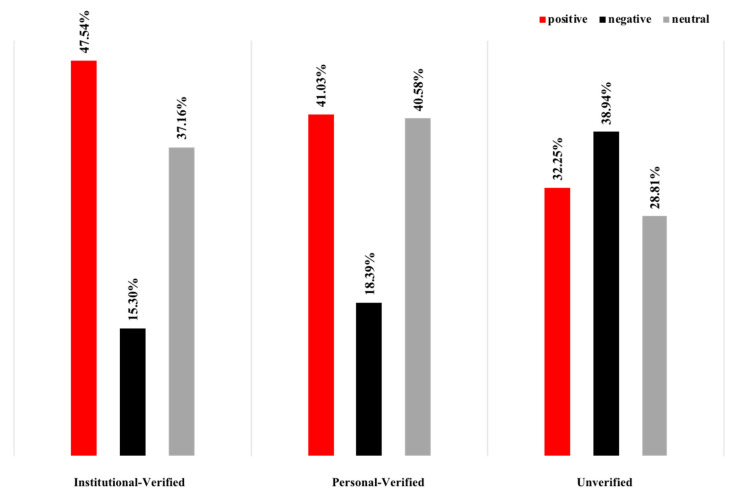
Emotional distribution by account type.

**Figure 10 ijerph-19-05197-f010:**
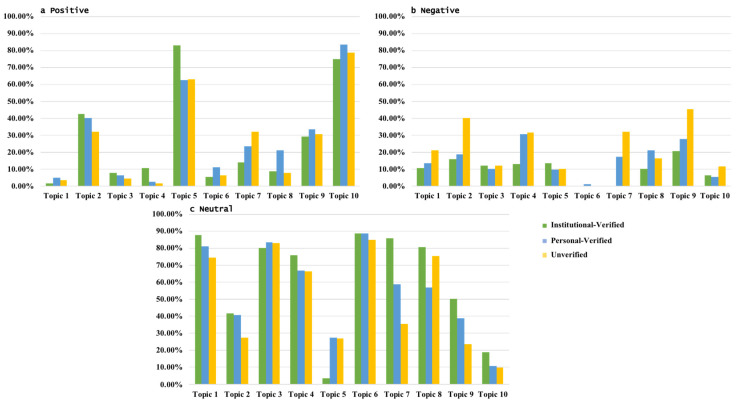
Emotion distribution by account type and focal topic.

**Figure 11 ijerph-19-05197-f011:**
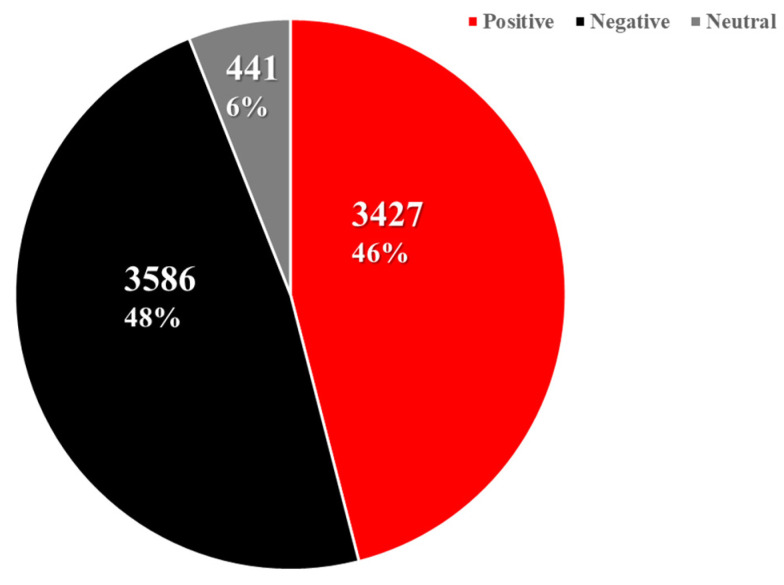
Emotional distribution of comments.

**Figure 12 ijerph-19-05197-f012:**
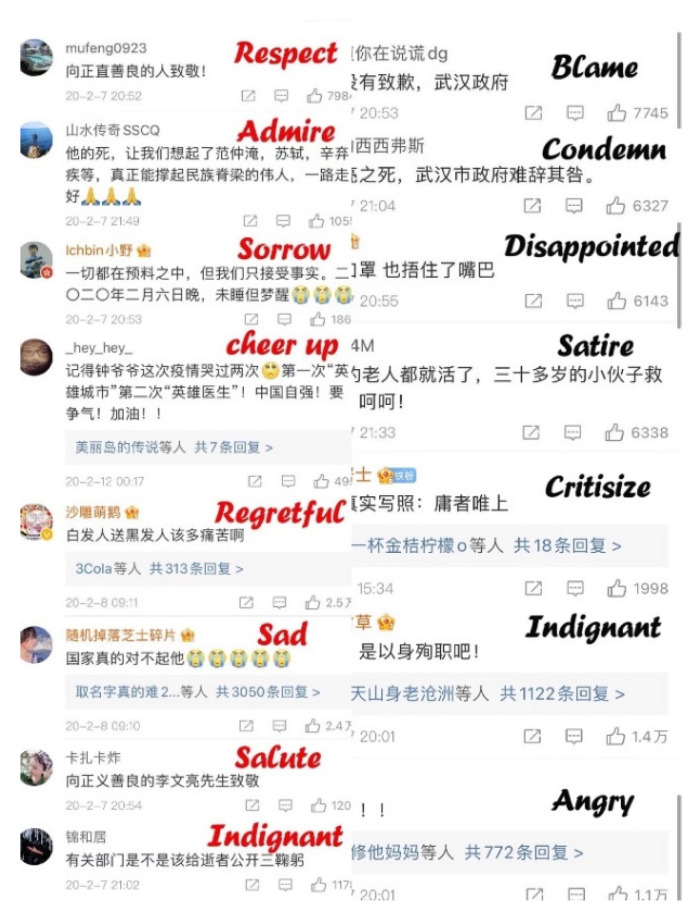
Positive and negative examples in comments.

**Figure 13 ijerph-19-05197-f013:**
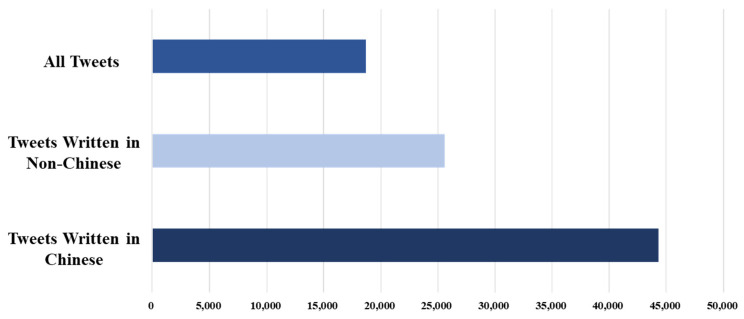
Tweets related to “Dr. Li Wenliang”.

**Figure 14 ijerph-19-05197-f014:**
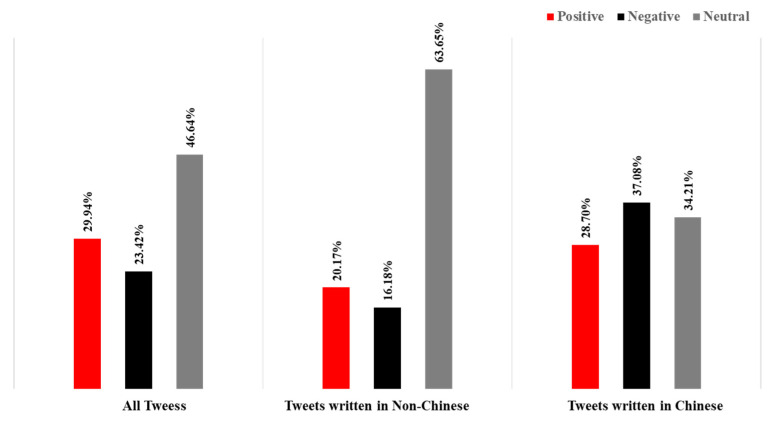
Emotional distribution by languages.

**Figure 15 ijerph-19-05197-f015:**
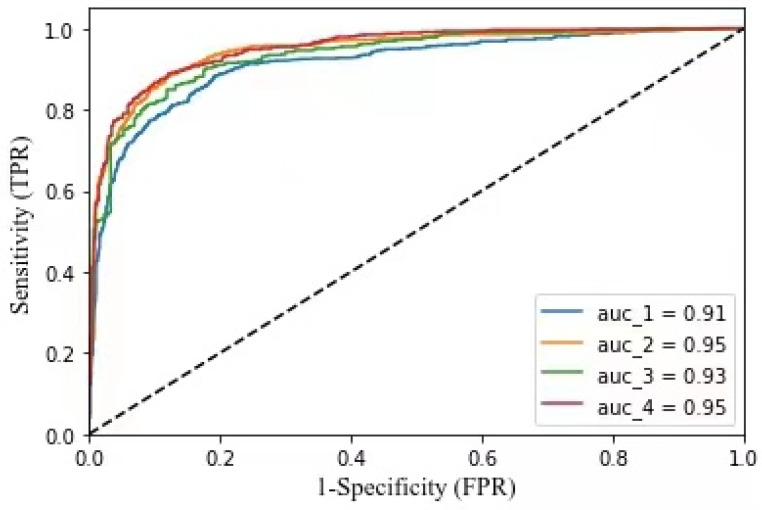
AUC diagrams of the algorithm.

**Figure 16 ijerph-19-05197-f016:**
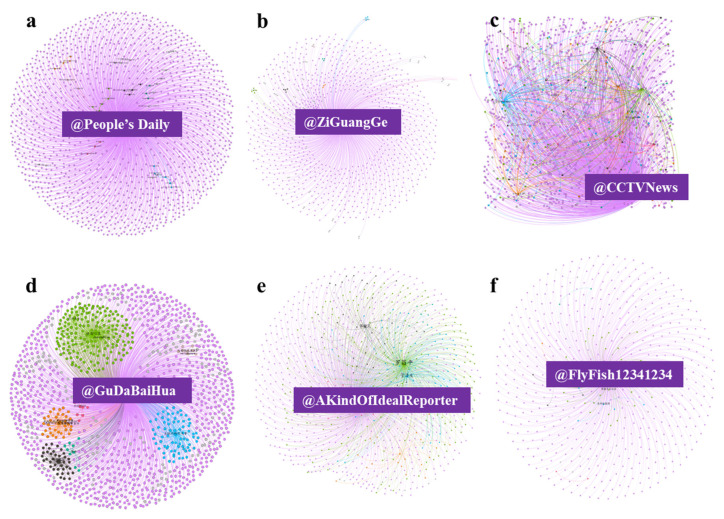
Network analysis of the various accounts.

**Table 1 ijerph-19-05197-t001:** Ten main focal topics related to the “Dr. Li Wenliang” incident.

No.	Date (2020)	Topic
1	1 January	Eight people who spread rumors regarding the pneumonia in Wuhan were investigated
2	29 January	Dr. Li Wenliang
3	29 January	Dialogue with doctors being admonished in Wuhan
4	29 January	Wuhan police reported the punishment of eight people for spreading untrue statements
5	29 January	Zeng Guang responded that eight people were interviewed in Wuhan
6	29 January	The Supreme Court talks about managing the COVID-19 rumors
7	1 February	Li Wenliang, a doctor from Wuhan, was admonished for diagnosing COVID-19
8	6 February	The State Supervision Commission sent an investigation team to investigate Dr. Li Wenliang
9	6 February	Dr. Li Wenliang passes away
10	12 February	Zhong Nanshan calls Li Wenliang a hero

**Table 2 ijerph-19-05197-t002:** Basic statistics for the related data.

Topic	Date (2020)	Num. of Messages	Num. of Forwards	Num. of Comments	Num. of Likes
Topic 1	1 January–1 March	1571	9584	15,401	97,442
Topic 2	29 January–10 February	151,543	1,539,146	1,387,729	13,457,656
Topic 3	29 January–19 February	6105	28,905	19,057	219,497
Topic 4	29 January–7 February	1034	45,805	16,511	205,367
Topic 5	29 January–1 March	16,499	37,379	19,669	291,677
Topic 6	29 January–10 February	92	933	1765	14,466
Topic 7	1 February–7 February	80	145	207	1112
Topic 8	6 February–1 March	688	24,342	12,410	124,471
Topic 9	6 February–1 March	9672	100,862	193,211	2,067,716
Topic 10	12 February–29 February	3342	5384	4771	64,210
Total	1 January–1 March	190,627	1,792,503	1,670,235	16,543,648

**Table 3 ijerph-19-05197-t003:** Statistics for the popular microblogs.

User_Name	Times	Interactive Comments	Likes	Positive	Negative	Neutral
@Toutiao News	2	391	10,044	44%	51%	4%
@Li Media	5	26,055	305,368	45%	49%	6%
@Pengpai News	6	1334	78,577	52%	43%	6%
@Beijing News	1	358	13,773	59%	38%	3%
@People Daily	2	40,804	460,541	35%	63%	2%
@People Concerned	1	394	3002	25%	64%	11%
@Vista News	1	175	5400	43%	50%	8%
@Phoenix News	3	3508	34,702	46%	46%	7%
@Fengmian News	1	505	2079	54%	39%	7%
@Guangming Daily	1	65	337	36%	59%	5%
@Universal Daily	1	409	4633	27%	68%	5%
@Souhu News	1	325	3940	42%	48%	10%
@Economic Daily	1	766	64,288	70%	24%	6%
@Xinhua News	1	120	623	75%	25%	0%
@Lanjing Economic	1	96	1087	94%	0%	6%
@CCTV News	1	53	1999	51%	46%	3%
@China Newsweek	2	2222	58,603	47%	47%	7%

**Table 4 ijerph-19-05197-t004:** Comparison of messages posted by the three account types with more than 500 forwards.

	Institutional-Verified Accounts	Personal-Verified Accounts	Unverified Accounts
	62% (181 of 291) 0.2%(181 of 77,015)	22% (64 of 291) 0.15%(64 of 42,515)	16% (46 of 291) 0.06%(46 of 71,097)
No. of accounts	92	54	42
Messages/No. of accounts	1.97	1.18	1.09
Original	162 (90%)	51 (80%)	36 (78%)
Video_True	69 (38%)	10 (16%)	6 (13%)
Picture_True	92 (51%)	35 (55%)	33 (72%)
Emoji_True	37 (20%)	17 (27%)	16 (35%)

**Table 5 ijerph-19-05197-t005:** Thematic analysis of the verified and unverified accounts.

Account Type	Theme
Institutional-Verified	**Theme 1.** *Dr. Li Wenliang from Wuhan Central Hospital was infected with COVID-19* **Theme 2.***Dr. Li Wenliang from Wuhan Central Hospital passed away* **Theme 3.***The state sends an investigation team to Wuhan*
Personal-Verified	**Theme 4.** *Dr. Li Wenliang is the “whistler”* **Theme 5.** *Dr. Li Wenliang received medical treatment* **Theme 6.** *The Wuhan government needs to make the information public*
Unverified	**Theme 7.***Dr. Li Wenliang was admonished for telling the truth* **Theme 8.** *The investigation team investigates the incident of Dr. Li Wenliang* **Theme 9.***Dr. Li Wenliang was admitted to ICU*

## Data Availability

By examing the event timelines and the associated hashtags on the popular Chinese social media site Sina-Weibo (https://weibo.com/login.php). The discussion about “Dr. Liwenliang” was taken as the research object and the event timeline and the Sina-Weibo tagging function focused on to analyze the behaviors and emotional changes in the social media users and elucidate the correlations.

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
