# Peer review of "Social Media User Behavior and Emotions during Crisis Events"

_ijerph, 2022, doi:10.3390/ijerph19095197_

Round 1

Reviewer 1 Report

An interesting and novel paper, especially in light of the pandemic context and the prominent use of social media platforms to distribute and share information in relation to Covid-19. However, the structure and use of sections throughout the paper is often unclear. Specific comments below:

The introduction is well structured and clear, with a strong rationale for the reason for this research. A few suggestions:

  • As I was reading the paper, I had noted that it would be worth providing a brief overview of the incident related to Dr Li Wenliang for readers who are not familiar with this. I discovered this was reported after the methods section on page 4. I believe this would be better placed in the introduction.
  • The text under the research questions (line 77 – 92) is better suited in a methods section.

Literature review

  • There appears to be some overlap in the content provided here and the introduction. I believe it would be clearer for the reader to merge the introduction and this section into one, minimising the repetition and building a clear argument. The research questions would be best suited at the end of this section.

Data collection and data synthesis:

  • The methods of data collection and synthesis are generally sound. However, the structure of this paper is unclear. In data collection and synthesis, the authors should report the methods of how the data was collected and subsequently synthesised. However, the authors appear to combine methods and results. For example, lines 243 – 244 and Table 2/Figure 4 should be placed in a clear results section. Similarly. Figure 5 and associated text written in the data analysis section should be in results, and lines 296 – 318 (incl. Figure 6 and 7) should be in the results section. There is also information in the data collection section that should be in the data synthesis section (e.g., lines 228 – 230).
  • It would be beneficial to clearly report the sections as data collection, data analysis and results – breaking the results down by the subheadings used in the data analysis section.
  • Not enough information is given on exactly how the analysis was conducted – i.e., how was the thematic analysis conducted (lines 482 – 487)? Did two or more authors analyse the data independently? Was this inductive or deductive analysis?

Conclusions and future research directions

  • There is very little exploration of findings here within the wider existing literature. It feels as though a discussion section with critical exploration of the current findings is missing.

Check grammar and punctuation throughout – e.g., line 234 ‘The first step is to eliminate invalid data’ and line 263 ‘the next step is emotional classification’ – these should be past tense. These are only a few examples so please check throughout.

Reviewer 2 Report

The abstract is clear and provides a wide view of the problematic approached in the article, as well as the main results.

The introduction is well structured, but lacks a better timeline (namely on COVID-19 abrupt start), as well as a more robust or clear reason of why the authors chose "Dr Li Wenliang" as the topic of interest in this work.  I suggest the authors to join the section "Literature review" with the "Introduction", and diminish the amount of detail about the methodology given in the introduction. 

The methodology used is interesting and innovating, and provides a realiable way to analyze social media comments. The context given to Dr Li Wenliang in the methods section might be partially included in the introduction, for the reader to be contextualized sooner in the reading. The authors need to further clarify ethical procedures: how was the privacy of the users assured? Was there any informed consent / request to use the data to the social media management team?

The results section needs work on turning the reading clearer. Figure 8 lacks an explanation, and its inclusion must be pondered, once it is only mentioned once and from the figure, without explanation, it is difficult to understand it. Were statistical analysis performed to assess differences between types of accounts / types of emotions or attitudes? If not, can they be performed?

Why was an analysis of twitter performed if this was not planned in the beggining, nor mentioned in the abstract?

The results section lacks organization. The authors should improve their clearness and parcimoniousness.

An overall review of the in text references should be performed. The references should appear in an alphabetical order, which sometimes do not happen (see, as an example, page 1, line 29 - Bastos et al., 2012 appears after Zubiaga et al., 2013). Additionally, there are places where the reference is not correctly done - e.g. page 1, line 36. I strongly suggest a throughout review of the english, namely to improve the clarification of sentences.

Reviewer 3 Report

It is interesting to research the behavior trends based on social media on the topic of Dr. Li Wen Liang. However, the figures used are not consistent, and can be hard to understand the significance of it. Suggestions to improve the presentation of the data. Additionally, the conclusion should have the average/mean of units/metrics indicating the significance of the findings. 

Round 2

Reviewer 1 Report

I thank the authors for addressing each query initially raised. I believe the manuscript is now clear and well-structured, reporting very interesting findings. I believe this is a timely and beneficial contribution to the field and I look forward to seeing it published in its present form.